# GPipe: Efficient Training of Giant Neural Networks using Pipeline Parallelism

**Yanping Huang, Youlong Cheng, Ankur Bapna, Orhan Firat, Mia Xu Chen, Dehao Chen, HyoukJoong Lee, Jiquan Ngiam, Quoc V. Le, Yonghui Wu, Zhifeng Chen**
{huangyp,ylc,ankurbpn,orhanf,miachen,dehao
hyouklee,jngiam,qvl,yonghui,zhifengc}
@google.com

## Abstract

Scaling up deep neural network capacity has been known as an effective approach to improving model quality for several different machine learning tasks. In many cases, increasing model capacity beyond the memory limit of a single accelerator has required developing special algorithms or infrastructure. These solutions are often architecture-specific and do not transfer to other tasks. To address the need for efficient and task-independent model parallelism, we introduce GPipe, a pipeline parallelism library that allows scaling any network that can be expressed as a sequence of layers. By pipelining different sub-sequences of layers on separate accelerators, GPipe provides the flexibility of scaling a variety of different networks to gigantic sizes efficiently. Moreover, GPipe utilizes a novel batch-splitting pipelining algorithm, resulting in almost linear speedup when a model is partitioned across multiple accelerators. We demonstrate the advantages of GPipe by training large-scale neural networks on two different tasks with distinct network architectures: (i) *Image Classification*: We train a 557-million-parameter AmoebaNet model and attain a top-1 accuracy of 84.4% on ImageNet-2012, (ii) *Multilingual Neural Machine Translation*: We train a single 6-billion-parameter, 128-layer Transformer model on a corpus spanning over 100 languages and achieve better quality than all bilingual models.

## 1 Introduction

Deep learning has seen great progress over the last decade, partially thanks to the development of methods that have facilitated scaling the effective capacity of neural networks. This trend has been most visible for image classification, as demonstrated by the accuracy improvements on ImageNet with the increase in model capacity (Figure 1a). A similar phenomenon can also be observed in the context of natural language processing (Figure 1b) where simple shallow models of sentence representations [1, 2] are outperformed by their deeper and larger counterparts [3, 4].

While larger models have brought remarkable quality improvements to several fields, scaling neural networks introduces significant practical challenges. Hardware constraints, including memory limitations and communication bandwidths on accelerators (GPU or TPU), force users to divide larger models into partitions and to assign different partitions to different accelerators. However, efficient model parallelism algorithms are extremely hard to design and implement, which often requires the practitioner to make difficult choices among scaling capacity, flexibility (or specificity to particular tasks and architectures) and training efficiency. As a result, most efficient model-parallel algorithms are architecture and task-specific. With the growing number of applications of deep learning, there is an ever-increasing demand for reliable and flexible infrastructure that allows researchers to easily scale neural networks for a large variety of machine learning tasks.

Figure 1: (a) Strong correlation between top-1 accuracy on ImageNet 2012 validation dataset [5] and model size for representative state-of-the-art image classification models in recent years [6, 7, 8, 9, 10, 11, 12]. There has been a $36\times$ increase in the model capacity. Red dot depicts $84.4\%$ top-1 accuracy for the 550M parameter AmoebaNet model. (b) Average improvement in translation quality (BLEU) compared against bilingual baselines on our massively multilingual in-house corpus, with increasing model size. Each point, $T(L, H, A)$, depicts the performance of a Transformer with $L$ encoder and $L$ decoder layers, a feed-forward hidden dimension of $H$ and $A$ attention heads. Red dot depicts the performance of a 128-layer 6B parameter Transformer.

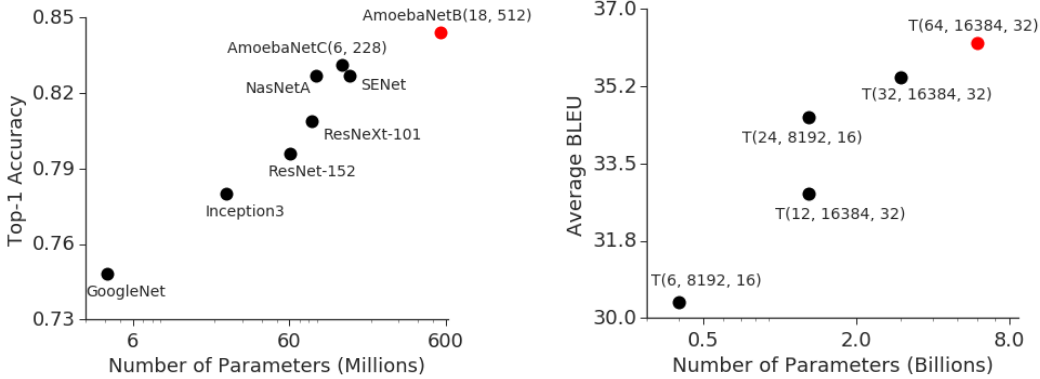

To address these challenges, we introduce GPipe, a flexible library that enables efficient training of large neural networks. GPipe allows scaling arbitrary deep neural network architectures beyond the memory limitations of a single accelerator by partitioning the model across different accelerators and supporting re-materialization on every accelerator [13, 14]. With GPipe, each model can be specified as a sequence of layers, and consecutive groups of layers can be partitioned into cells. Each cell is then placed on a separate accelerator. Based on this partitioned setup, we propose a novel pipeline parallelism algorithm with batch splitting. We first split a *mini-batch* of training examples into smaller *micro-batches*, then pipeline the execution of each set of micro-batches over cells. We apply synchronous mini-batch gradient descent for training, where gradients are accumulated across all micro-batches in a mini-batch and applied at the end of a mini-batch. Consequently, gradient updates using GPipe are consistent regardless of the number of partitions, allowing researchers to easily train increasingly large models by deploying more accelerators. GPipe can also be complemented with data parallelism to further scale training.

We demonstrate the flexibility and efficiency of GPipe on image classification and machine translation. For image classification, we train the AmoebaNet model on $480 \times 480$ input from the ImageNet 2012 dataset. By increasing the model width, we scale up the number of parameters to $557$ million and achieve a top-1 validation accuracy of $84.4\%$. On machine translation, we train a single 128-layer 6-billion-parameter multilingual Transformer model on 103 languages (102 languages to English). We show that this model is capable of outperforming the individually trained 350-million-parameter bilingual Transformer Big [15] models on all 102 language pairs.

## 2   The GPipe Library

We now describe the interface and the main design features of GPipe. This open-source library is implemented under the Lingvo [16] framework. The core design features of GPipe are generally applicable and can be implemented for other frameworks [17, 18, 19].

### 2.1   Interface

Any deep neural network can be defined as a sequence of $L$ layers. Each layer $L_i$ is composed of a forward computation function $f_i$, and a corresponding set of parameters $w_i$. GPipe additionally allows the user to specify an optional computation cost estimation function, $c_i$. With a given number of partitions $K$, the sequence of $L$ layers can be partitioned into $K$ composite layers, or cells. Let $p_k$ consist of consecutive layers between layers $i$ and $j$. The set of parameters corresponding to $p_k$ is

Figure 2: (a) An example neural network with sequential layers is partitioned across four accelerators. $F_k$ is the composite forward computation function of the $k$-th cell. $B_k$ is the back-propagation function, which depends on both $B_{k+1}$ from the upper layer and $F_k$. (b) The naive model parallelism strategy leads to severe under-utilization due to the sequential dependency of the network. (c) Pipeline parallelism divides the input mini-batch into smaller micro-batches, enabling different accelerators to work on different micro-batches simultaneously. Gradients are applied synchronously at the end.

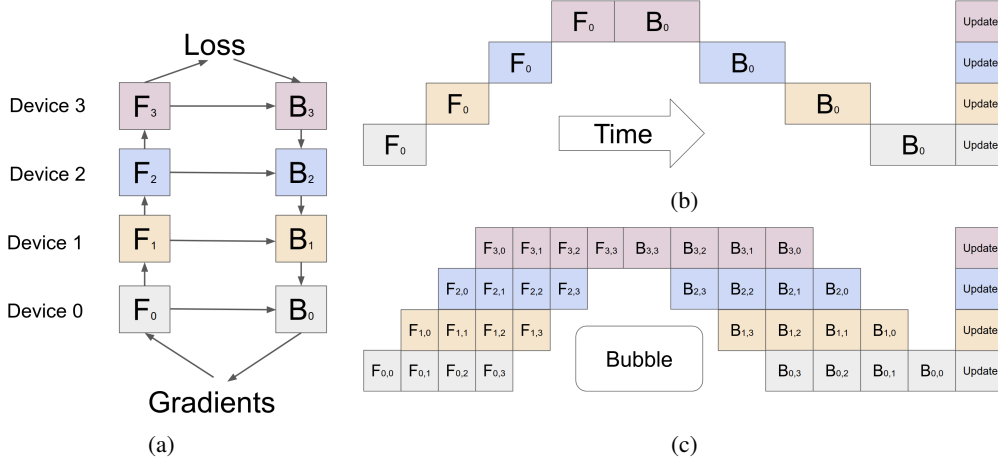

equivalent to the union of $w_i, w_{i+1}, \ldots, w_j$, and its forward function would be $F_k = f_j \circ \ldots \circ f_{i+1} \circ f_i$. The corresponding back-propagation function $B_k$ can be computed from $F_k$ using automatic symbolic differentiation. The cost estimator, $C_k$, is set to $\Sigma_{l=i}^{j} c_l$.

The GPipe interface is extremely simple and intuitive, requiring the user to specify: (i) the number of model partitions $K$, (ii) the number of micro-batches $M$, and (iii) the sequence and definitions of $L$ layers that define the model. Please refer to supplementary material for examples.

## 2.2 Algorithm

Once the user defines the sequence of layers in their network in terms of model parameters $w_i$, forward computation function $f_i$, and the cost estimation function $c_i$, GPipe partitions the network into $K$ cells and places the $k$-th cell on the $k$-th accelerator. Communication primitives are automatically inserted at partition boundaries to allow data transfer between neighboring partitions. The partitioning algorithm minimizes the variance in the estimated costs of all cells in order to maximize the efficiency of the pipeline by syncing the computation time across all partitions.

During the forward pass, GPipe first divides every mini-batch of size $N$ into $M$ equal micro-batches, which are pipelined through the $K$ accelerators. During the backward pass, gradients for each micro-batch are computed based on the same model parameters used for the forward pass. At the end of each mini-batch, gradients from all $M$ micro-batches are accumulated and applied to update the model parameters across all accelerators. This sequence of operations is illustrated in Figure 2c.

If batch normalization [20] is used in the network, the sufficient statistics of inputs during training are computed over each *micro-batch* and over replicas if necessary [21]. We also track the moving average of the sufficient statistics over the entire *mini-batch* to be used during evaluation.

## 2.3 Performance Optimization

In order to reduce activation memory requirements, GPipe supports re-materialization [14]. During forward computation, each accelerator only stores output activations at the partition boundaries. During the backward pass, the $k$-th accelerator recomputes the composite forward function $F_k$. As a consequence, peak activation memory requirement is reduced to $O(N + \frac{L}{K} \times \frac{N}{M})$, where $\frac{N}{M}$ is the micro-batch size and $\frac{L}{K}$ is the number of layers per partition. In comparison, memory requirement without re-materialization and partitioning would be $O(N \times L)$, since computing the gradients $b_i$ requires both the upper layer gradients $b_{i+1}$ and the cached activations $f_i(x)$.

Table 1: Maximum model size of AmoebaNet supported by GPipe under different scenarios. Naive-1 refers to the sequential version without GPipe. Pipeline-$k$ means $k$ partitions with GPipe on $k$ accelerators. AmoebaNet-D (L, D): AmoebaNet model with $L$ normal cell layers and filter size $D$. Transformer-L: Transformer model with $L$ layers, 2048 model and 8192 hidden dimensions. Each model parameter needs 12 bytes since we applied RMSProp during training.

| NVIDIA GPUs (8GB each) | Naive-1 | Pipeline-1 | Pipeline-2 | Pipeline-4 | Pipeline-8 |
|---|---|---|---|---|---|
| AmoebaNet-D (L, D) | (18, 208) | (18, 416) | (18, 544) | (36, 544) | (72, 512) |
| # of Model Parameters | 82M | 318M | 542M | 1.05B | 1.8B |
| Total Model Parameter Memory | 1.05GB | 3.8GB | 6.45GB | 12.53GB | 24.62GB |
| Peak Activation Memory | 6.26GB | 3.46GB | 8.11GB | 15.21GB | 26.24GB |
| Cloud TPUv3 (16GB each) | Naive-1 | Pipeline-1 | Pipeline-8 | Pipeline-32 | Pipeline-128 |
| Transformer-L | 3 | 13 | 103 | 415 | 1663 |
| # of Model Parameters | 282.2M | 785.8M | 5.3B | 21.0B | 83.9B |
| Total Model Parameter Memory | 11.7G | 8.8G | 59.5G | 235.1G | 937.9G |
| Peak Activation Memory | 3.15G | 6.4G | 50.9G | 199.9G | 796.1G |

As illustrated in Figure 2c, partitioning introduces some idle time per accelerator, which we refer to as the *bubble* overhead. This bubble time is $O(\frac{K-1}{M+K-1})$ amortized over the number of micro-steps $M$. In our experiments, we found the bubble overhead to be negligible when $M \geq 4 \times K$. This is also partly because re-computation during the backward pass can be scheduled earlier, without waiting for the gradients from earlier layers.

GPipe also introduces low communication overhead, given that we only need to pass activation tensors at the partition boundaries between accelerators. Therefore, we can achieve efficient scaling performance even on accelerators without high-speed interconnects.

Figure 2c assumes partitions are evenly balanced. However, memory requirements and computation flops at different layers are often quite imbalanced. In such scenarios, imperfect partitioning algorithms might lead to load imbalance. Better partitioning algorithms can potentially improve the performance over our heuristic approach.

# 3 Performance Analyses

We evaluate GPipe performance with two very different types of model architectures: an AmoebaNet [12] convolutional model and a Transformer [15] sequence-to-sequence model. We ran experiments to study their scalability, efficiency and communication cost.

We expect both re-materialization and pipeline parallelism to benefit memory utilization and thus make fitting giant models feasible. We report the biggest model size GPipe can support under reasonably large input size in Table 1. For AmoebaNet, we ran the experiments on Cloud TPUv2s with 8GB memory per accelerator. We used a fixed input image size of $224 \times 224$ and mini-batch size of $128$. Without GPipe, a single accelerator can train up to an 82M-parameter AmoebaNet, constrained by device memory limits. Owing to re-materialization in back-propagation and batch splitting, GPipe reduces the intermediate activation memory requirements from 6.26GB to 3.46GB, enabling a 318M-parameter model on a single accelerator. With model parallelism, we were able to scale AmoebaNet to 1.8 billion parameters on 8 accelerators, 25x more than what is possible without GPipe. In this case, the maximum model size did not scale perfectly linearly due to the imbalanced distribution of model parameters over different layers in AmoebaNet.

We next trained Transformer models using Cloud TPUv3s with 16GB memory per accelerator core. We used a fixed vocabulary size of 32k, sequence length 1024 and batch size 32. Each Transformer layer has 2048 for model dimension, 8192 for feed-forward hidden dimension and 32 attention heads. We scaled the model by varying the number of layers. Re-materialization allows training a $2.7\times$ larger model on a single accelerator. With 128 partitions, GPipe allows scaling Transformer up to 83.9B parameters, a $298\times$ increase than what is possible on a single accelerator. Different from AmoebaNet, the maximum model size scales linearly with the number of accelerators for Transformer, since each layer has the same number of parameters and input sizes.

To evaluate efficiency, we report the normalized training throughput of AmoebaNet-D (18, 256) and Transformer-48 using GPipe with different numbers of partitions and different numbers of micro-batches in Table 2. Each partition is assigned to a separate accelerator. We observe that when the number of micro-batches $M$ is at least $4\times$ the number of partitions, the bubble overhead is almost negligible. For Transformer model, there is a $3.5\times$ speedup when it is partitioned across four times more accelerators. Furthermore, training throughput scales almost linearly with the number of devices, thanks to the computation being evenly distributed across Transformer layers. In contrast, the AmoebaNet

Table 2: Normalized training throughput using GPipe with different # of partitions $K$ and different # of micro-batches $M$ on TPUs. Performance increases with more micro-batches. There is an almost linear speedup with the number of accelerators for Transformer model when $M \gg K$. Batch size was adjusted to fit memory if necessary.

| TPU | AmoebaNet | | | Transformer | | |
|---|---|---|---|---|---|---|
| $K =$ | 2 | 4 | 8 | 2 | 4 | 8 |
| $M = 1$ | 1 | 1.13 | 1.38 | 1 | 1.07 | 1.3 |
| $M = 4$ | 1.07 | 1.26 | 1.72 | 1.7 | 3.2 | 4.8 |
| $M = 32$ | 1.21 | 1.84 | 3.48 | 1.8 | 3.4 | 6.3 |

model achieves sub-linear speedup due to its imbalanced computation distribution. When $M$ is relatively small, the bubble overhead can no longer be negligible. When $M$ is 1, there is effectively no pipeline parallelism. We observe relatively constant throughput regardless of the number of accelerators used, indicating only one device is actively computing at any given time.

To measure the effect of communication overhead with GPipe, we ran our experiments on a single host with multiple NVIDIA P100 GPUs but without NVLinks. Data transfer across GPUs then has to involve the relatively slow device-to-host and host-to-device transfers through PCI-E. The number of micro-batches was fixed at 32. As shown in Table 3, we observe $2.7\times$ speedup for AmoebaNet-D (18, 128) when we increase the number of partitions from 2 to 8. For the 24-layer Transformer,

the speedup is $3.3\times$. There is similar linear speedup to what we observe on TPUs where high-speed interconnects are equipped. The communication bandwidth between devices is no longer a bottleneck for model parallelism since GPipe only transfers activation tensors at the boundaries of partitions.

Table 3: Normalized training throughput using GPipe on GPUs without high-speed interconnect.

| GPU | AmoebaNet | | | Transformer | | |
|---|---|---|---|---|---|---|
| $K =$ | 2 | 4 | 8 | 2 | 4 | 8 |
| $M = 32$ | 1 | 1.7 | 2.7 | 1 | 1.8 | 3.3 |

## 3.1 Performance Overhead Breakdown

To study opportunities for future performance improvements, we identified the key factors that affect the performance of GPipe on Cloud TPUs. We measured the time spent on different activities listed in Table 4. We found that re-computation time was the main contributor to GPipe overhead, taking up to $23\%$ of the total step time. Another source of overhead was load imbalance. With two partitions, overhead caused by load imbalance was only $3.2\%$. The theoretical bubble overhead is $O\left(\frac{K-1}{M+K-1}\right)$ where $K$ is the number of partitions and $M$ is the number of micro-batches in each mini-batch. The observed bubble overhead was slightly lower than the theoretical value partly

Table 4: Time step breakdown

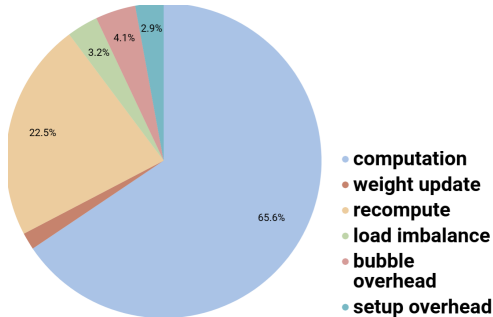

because re-computation was scheduled early to overlap with the bubble. Weight update time for gradient aggregation at the end of pipeline was also small, thanks to high-speed interconnections between the accelerators.

## 4  Image Classification

As a proof of concept, we first used GPipe to scale AmoebaNet. We increased the number of channels in an AmoebaNet and scaled the input image size to $480 \times 480$. We trained this 557-million-parameter AmoebaNet-B(18, 512) on the ImageNet 2012 dataset, using the same hyper-parameters as described

Table 5: Image classification accuracy using AmoebaNet-B (18, 512) first trained on ImageNet 2012 then fine-tuned on others. Please refer to the supplementary material for a detailed description of our training setup. Our fine-tuned results were averaged across 5 fine-tuning runs. Baseline results from Real *et al.* [12] and Cubuk *et al.* [26] were directly trained from scratch. *Mahajan *et al.*'s model [27] achieved $85.4\%$ top-1 accuracy but it was pretrained on non-public Instagram data. Ngiam *et al.* [28] achieved better results by pre-training with data from a private dataset (JFT-300M).

| Dataset | # Train | # Test | # Classes | Accuracy (%) | Previous Best (%) |
|---|---|---|---|---|---|
| ImageNet-2012 | 1,281,167 | 50,000 | 1000 | **84.4** | 83.9 [12] (85.4*[27]) |
| CIFAR-10 | 50,000 | 10,000 | 10 | **99.0** | 98.5 [26] |
| CIFAR-100 | 50,000 | 10,000 | 100 | **91.3** | 89.3 [26] |
| Stanford Cars | 8,144 | 8,041 | 196 | 94.6 | **94.8**$^*$ [26] |
| Oxford Pets | 3,680 | 3,369 | 37 | **95.9** | 93.8$^*$ [29] |
| Food-101 | 75,750 | 25,250 | 101 | **93.0** | 90.4$^*$ [30] |
| FGVC Aircraft | 6,667 | 3,333 | 100 | 92.7 | **92.9**$^*$ [31] |
| Birdsnap | 47,386 | 2,443 | 500 | **83.6** | 80.2$^*$ [32] |

in [12]. The network was divided into 4 partitions. This single model achieves $84.4\%$ top-1 and $97\%$ top-5 validation accuracy with single-crop.

We further demonstrate the effectiveness of giant convolution networks on other image datasets through transfer learning [22, 23]. Specifically, we used the pre-trained ImageNet model to fine-tune on a variety of target datasets ranging from general to fine-grained classification. We changed the number of output units in the last softmax classification layer to the number of classes in the target dataset and initialized the new softmax layer randomly. All the other layers were initialized from ImageNet pre-training. Input images to the network during training were resized to $480 \times 480$, horizontally flipped randomly and augmented using cutout [24]. Training hyper-parameters were the same as those used for ImageNet (a detailed description of our training setup is provided in supplementary material). In Table 5, we report the average single-crop test accuracy over 5 fine-tuning runs for each dataset. Our giant models obtain competitive results on all target datasets. For example, CIFAR-10 error rate is reduced to $1\%$ and CIFAR-100 error rate to $8.7\%$. These results corroborate the findings by Kornblith *et al.* [25], i.e., better ImageNet models transfer better.

## 5 Massive Massively Multilingual Machine Translation

Next, we demonstrate the flexibility of GPipe by scaling up models used for Natural Language Processing (NLP). Due to an abundance of available parallel corpora, neural machine translation (NMT) has become a benchmark task for any architecture used for NLP [33, 15, 34, 35, 36]. For this reason, we continue our GPipe experiments on a large-scale multilingual NMT task. We use a corpus of parallel documents over 102 languages and English, containing a total of 25 billion training examples, ranging from $10^4$ to $10^9$ per language [37]. This dataset creates a realistic test bed for experiments on scalability by spanning a diverse set of languages from data-scarce (low-resource) to data-rich (high-resource). For the first time in machine translation, we show that a large enough NMT model can learn the mapping between more than 100 language pairs simultaneously, while achieving better than bilingual model performance for all languages. This further brings out the importance of having efficient and flexible model-parallelism tools.

Our comparison is based on the performance of a single Transformer [15] trained on all language pairs in this corpus. We scale the architecture along two dimensions to stress the flexibility of GPipe: (i) along the depth by increasing the number of layers in the model and (ii) along the width by increasing the hidden dimension in the feed-forward layers and the number of attention heads (as well as # attention channels) in multi-head attention layers similar to Shazeer *et al.* [34]. Please refer to the supplementary material for a detailed description of our dataset, baselines, training configuration and optimization hyper-parameters.

We start with a standard 400M-parameter Transformer Big model, $T(6, 8192, 16)^1$, as described in Chen *et al.* [35], with a vocabulary size of 64k. In Figure 3, we compare its performance against a

---

$^1T(L, H, A)$ is a Transformer model with $L$ encoder layers and $L$ decoder layers, a feed-forward hidden dimension of $H$ and $A$ attention heads. The model dimension is fixed to 1024.

1.3B-parameter deep model, $T(24, 8192, 16)$, a 1.3B-parameter wide model, $T(12, 16384, 32)$, a 3B-parameter model, $T(32, 16384, 32)$ and a 6B-parameter model, $T(64, 16384, 32)$. All of the models are trained on all language pairs simultaneously, using temperature-based sampling as employed for multilingual BERT[2] [3]. $T(12, 16384, 32)$, $T(24, 8192, 32)$, $T(32, 16384, 32)$ and $T(64, 16384, 32)$ are partitioned over 2, 4, 8 and 16 accelerators respectively.

From Figure 3, we can observe that increasing the model capacity from 400M to 1.3B parameters significantly improves performance across all languages. Scaling up the model from 1.3B parameters to 6B parameters shows further improvement, especially for high-resource languages. Below we discuss some of our empirical findings based on these large-scale experiments.

Figure 3: Translation quality across all languages with increasing multilingual model capacity. Languages are arranged in the order of decreasing training dataset size from left to right. $T(L, H, A)$, depicts the performance of a Transformer with $L$ encoder and $L$ decoder layers, a feed-forward hidden dimension of $H$ and $A$ attention heads. We notice that increasing the model capacity, from 400M params ($T(6, 8192, 16)$) to 1.3B ($T(24, 8192, 16)$), and further, to 6B ($T(64, 16384, 32)$), leads to significant quality improvements across all languages. We also notice huge quality improvements for low-resource languages (right side of the plot), when compared against bilingual baselines, highlighting the significant transfer gains resulting from training a multilingual model.

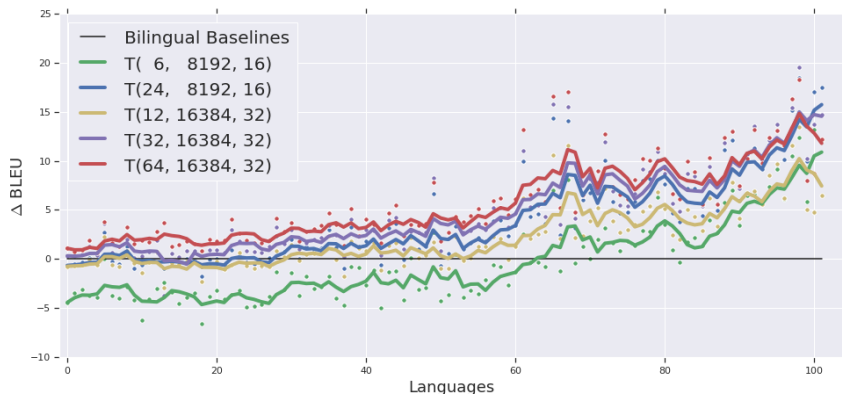

**Depth-Width Trade-off:** We study the trade-off between depth and width in our multilingual setup and compare the performance of 1.3B wide model $T(12, 16384, 32)$ and 1.3B deep model $T(24, 8192, 16)$. While the quality of these two models on high-resource languages (left of Figure 3) is very similar, the deeper model outperforms by huge margins on low-resource languages, suggesting that increasing model depth might be better for generalization. Further, the quality improvements for low-resource languages (right side of Figure 3), when comparing the 1.3B deep model against the 400M model, are almost as large as the improvements for high-resource languages, indicating that increasing depth might potentially increase the extent of transfer to low-resource tasks.

**Trainability Challenges with Deep Models:** Although depth increases the representational capacity of neural networks, it also complicates the optimization problem. In our large-scale experiments, we encountered severe trainability issues arising from a combination of sharp activations (positive kurtosis) and dataset noise. We observed that after training for a few thousand steps, the model predictions would become extremely peaky and vulnerable to noise, which frequently resulted in non-finite or large gradients that eventually destroyed the learning progress. To counter these problems, we apply two methods: (i) Following Zhang *et al.* [38], we scale down the initialization of all transformer feed-forward layers by the number of layers. (ii) We clip the logit predictions (softmax pre-activations) whenever their magnitude exceeds a certain value. A combination of these two approaches allows us to mitigate the training instability posed by scaling model depth.

# 6   Design Features and Trade-Offs

Several approaches have been proposed to enable efficient large-scale model parallelism. However, each approach chooses its own set of trade-offs, making it suitable for scaling specific architectures

under particular hardware constraints. The core idea of model parallelism involves partitioning a network into different computational units, which are then placed on different devices [39, 40, 41, 42]. Conceptually this supports scaling a large spectrum of models to huge capacities. However these approaches typically suffer from low hardware utilization and communication bottlenecks. Single Program Multiple Data (SPMD) and pipeline parallelism have been proposed as solutions to counter these challenges.

Mesh-Tensorflow [34] follows the SPMD paradigm, which extends the Single Instruction Multiple Data (SIMD) approach used for data parallelism to other tensor dimensions. SPMD allows splitting every computation across multiple devices, allowing the user to scale the size of individual matrix multiplications (and thus, the model parameters of individual layers) linearly with the number of accelerators. However, this also introduces high communication overhead between the accelerators due to an abundance of AllReduce-like operations used to combine the outputs of each parallelized matrix multiplication. This limits the applicability of the approach to scenarios where accelerators are connected with high speed interconnects. Further, SPMD limits the type of operations that can be efficiently scaled, restricting its use to a specific set of network architectures and machine learning tasks. For example, splitting along the channel dimension of convolution layers under this paradigm is not efficient given that channels are effectively fully connected, whereas splitting along the spatial dimension requires sophisticated techniques for the halo regions. While SPMD allows scaling the model depth by making each operation smaller, it requires splitting each layer over a larger number of accelerators, which in turn further increases the communication overhead across devices.

Other approaches have attempted to utilize pipeline-parallelism-based approaches to scale neural networks [43, 44]. The most recent iteration of pipeline parallelism applied to neural network training is PipeDream [45], which targets reducing the communication overhead for parameter servers [46]. PipeDream pipelines the execution of forward passes and intersperses them with backward passes in an attempt to maximize hardware utilization. This design suffers from weight staleness introduced by asynchronous backward updates. To avoid optimization issues stemming from the weight staleness, PipeDream requires maintaining multiple versioned copies of the model parameters on each accelerator in order to compute the gradient updates accurately, preventing users from scaling to bigger models.

GPipe introduces a new brand of pipeline parallelism that pipelines the execution of *micro-batches* before applying a single synchronous gradient update for the entire *mini-batch*. Our novel batch-splitting pipeline parallelism algorithm, when combined with re-materialization, allows scaling to a large number of micro-batches. This minimizes the *bubble* overhead without the need for asynchronous gradient updates. GPipe enables the user to scale model size linearly with the number of accelerators used. Unlike SPMD, pipeline parallelism introduces little additional communication overhead when scaling the model. Inter-device communication only takes place at partition boundaries for every micro-batch and the introduced communication overhead is marginal, extending the utility of GPipe to situations where high-speed device interconnects are not available. However, GPipe currently assumes that a single layer fits within the memory requirements of a single accelerator[3]. Additionally, micro-batch splitting requires complicated strategies to support layers that require computations across the batch (for example, BatchNorm uses statistics over the micro-batch during training, but accumulates mini-batch statistics for evaluation).

## 7    Conclusion

In this work, we introduce GPipe, a scalable model-parallelism library for training giant networks. We propose a novel batch-splitting pipeline-parallelism algorithm that uses synchronous gradient updates, allowing model parallelism with high hardware utilization and training stability. We leverage GPipe to train large-scale convolutional and transformer-based models and demonstrate strong empirical results on both image classification and multilingual machine translation. We highlight three key attributes of GPipe: 1) Efficiency: Using a novel batch-splitting pipelining algorithm, GPipe achieves almost linear speedup with the number of devices. 2) Flexibility: GPipe supports any sequential neural networks. 3) Reliability: GPipe utilizes synchronous gradient descent and guarantees consistent training regardless of the number of partitions.

## Footnotes

[2]https://github.com/google-research/bert/blob/master/multilingual.md

[3]One possible way around this limitation is splitting a single matrix-multiplication into smaller ones and spreading them sequentially across multiple layers.

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
