[Supplementary Material]

# Supplementary Material for GPipe

## 1 GPipe Example Usage

Users of GPipe library first express their neural network as a sequential list of $L$ layers. Any computational graph can be partitioned into sequence of sub-graphs. Example basis layers include convolution, pooling, batch normalization, dropout, transformer, softmax and others. Layers that are connected sequentially or in parallel can be combined into a new composite layer. Users can combine any number of layers in arbitrary ways as long as the composite forward function is properly defined.

Figure 1 shows an example use case of the GPipe library. It is a unit test for consistent training that verifies the norms of all gradients in this example network to be the same within numerical errors, regardless of the number of partitions.

## 2 Image Classification Training Details

### 2.1 Training Hyperparameters

We trained an AmoebaNet-B (18, 512) with $557$ million model parameters and input image size of $480 \times 480$ on the ImageNet ILSVRC-2012 dataset. We followed the same hyperparameters and input preprocessing as described in [2] to train AmoebaNet-B (18, 512). We employed the RMSProp optimizer with a decay of $0.9$ and $\epsilon = 0.1$, $L^2$ regularization $\lambda = 4 \times 10^{-5}$, label smoothing coefficiency $0.1$ and an auxiliary head with weight $0.4$. We applied the same drop-path schedule to intermediate layers as in NasNet [3] and dropout to the final layer with probability $0.5$. We used a learning rate schedule that decays every 3 epochs at a rate of $0.97$ with an initial learning rate of $0.00125$ times the batch size.

In Table 1, we reported the hyperparameters used for each transfer learning dataset. We selected learning rate from the set $\{0.0125, 0.00375, 0.075, 0.115, 0.15\}$ and $L^2$ weight decay from the set $\{0, 4e-8, 4e-7, 4e-6, 4e-5\}$. The selection was based on a hold-out subset ($20\%$) of training data. We then applied the selected hyperparameters for the final training and repeated five times.

| Dataset | Learning Rate | $L^2$ Weight Decay |
|---|---|---|
| CIFAR-10 | 0.0125 | 4e-05 |
| CIFAR-100 | 0.0125 | 4e-5 |
| Stanford Cars | 0.15 | 0 |
| Oxford-IIIT Pets | 0.00375 | 4e-06 |
| Food-101 | 0.15 | 4e-08 |
| FGVC Aircraft | 0.075 | 4e-08 |
| Birdsnap | 0.15 | 4e-06 |

Table 1: Hyperparameters used in transfer learning. We selected the learning rate and $L^2$ weight regularization parameters for each dataset on a hold-out subset of training dataset. For other hyperparameters we used the same ones as in ImageNet training.

Figure 1: Sample unit test under TensorFlow Lingvo [1] to ensure that GPipe provides consistent training regardless of the number of partitions.

```python
import tensorflow as tf

from lingvo.core import base_layer
from lingvo.core import layers
from lingvo.core import py_utils
from TensorPipe import TensorPipeLayer

def BuildDummyTensorPipeLayer(
    num_layers=16,
    num_splits=4,
    num_mirco_batches=8):
  assert num_layers % num_splits == 0
  layers = []
  # Construct a dummy layer with 16 3x3 conv layers
  for i in range(num_layers):
    layers.append(layers.Conv2DLayer.Params().Set(
        name='layer_{}'.format(i)),
        filter_shape=(3, 3, 1, 1),
        filter_stride=(1, 1))
  # Evenly distribute layers to partitions.
  partitions = []
  layers_per_split = num_layers // num_splits
  for split in range(num_splits):
    sub = layers[
        split * layers_per_split:
        (split + 1) * layers_per_split]
    partitions.append(sub)
  # Build pipeline parallelism model using TensorPipe
  p = TensorPipeLayer.Params().Set(
        name='TensorPipe',
        num_mirco_batches=num_mirco_batches,
        partitions=partitions)
  layer = p.cls(p)
  return layer

class DummyTensorPipeTest(tf.test.TestCase):

  def _verify_consistent_training(self, num_splits):
    g = tf.Graph()
    with g.as_default():
      py_utils.GetOrCreateGlobalStep()
      tf.set_random_seed(88888888)
      inputs = tf.random_uniform([16, 8, 8, 1])
      net = BuildDummyTensorPipeLayer(num_splits=num_splits)
      logits = net.FPropDefaultTheta(inputs)
      loss = tf.reduce_mean(logits)
      grads = tf.gradients(
          loss, tf.trainable_variables())
      grad_norm = tf.sqrt(
          py_utils.SumSquared(grads))
    with self.session(graph=g) as sess:
      sess.run(tf.global_variables_initializer())
      grad_norm_val, = sess.run([grad_norm])
      # Verify grad norm is the same regardless of
      # the number of splits
      self.assertNear(
          grad_norm_val, 0.269997, err=1.0e-6)

  def testDummyPipelineCnnOneSplit(self):
    self._verify_timestep_counts(num_splits=1)

  def testDummyPipelineCnnTwoSplits(self):
    self._verify_timestep_counts(num_splits=2)

  def testDummyPipelineCnnFourSplits(self):
    self._verify_timestep_counts(num_splits=4)
```

## 2.2 Consistent Training

GPipe performs synchronous training over the micro-batches. In this section, we verified the hypothesis that the end-to-end convergence accuracy using GPipe is the same within statistical errors, regardless of the number of partitions. We trained AmoebaNet-D (6, 256) several times for 35 epochs and measured the final validation accuracy on ImageNet. We chose AmoebaNet-D (6, 256) since it was the winning image model by training cost in the DAWNBench competition [4]. We adopted the same hyperparameters and training procedure reported in DAWNBench.[1] As a baseline, we trained AmoebaNet-D (6, 256) 5 times using the official open-source implementation and computed the mean and standard deviation of the final accuracy. Using the same hyperparameters and training procedures, we trained the same network using GPipe with 1, 2, 4 and 8 partitions. We found that the resulting accuracy fell within 1.6 standard deviations from the mean, as expected.

# 3 Machine Translation Training Details

We study multilingual NMT on a massive scale, using a corpus generated by crawling and extracting parallel sentences from the web. Figure 2 illustrates the data distribution across languages for all 102 languages studied in this paper.

Figure 2: Per language pair data distribution of the dataset used for our multilingual experiments. The y-axis depicts the number of training examples available per language on a logarithmic scale. Dataset sizes range from 35k for the lowest-resource language to 2 billion for the highest.

## 3.1 Baselines

For our bilingual experiments, we used variants of the Transformer architecture [5]. For most bilingual experiments, we used a larger version of Transformer Big model containing 375M parameters, and a shared source-target SPM vocabulary with 32k tokens. We tuned different values of dropout, depending on the dataset size for each language pair. For most medium and low-resource languages we also experimented with Transformer Base model. All our models were trained with Adafactor [6] with momentum factorization, a learning rate schedule of (3.0, 40K),[2] and a per-parameter norm clipping threshold of 1.0. For Transformer Base models, we used a learning rate schedule of (2.0, 8K). BLEU scores were computed using the checkpoint with the best validation performance on true-cased output and references.[3]

## 3.2 Multilingual Baselines

We now describe our approach for training the multilingual models. Due to the large imbalance in our training dataset (Figure 2), we first designed a sampling strategy to simultaneously train a single model on over 200 language pairs. Sampling directly from the data distribution would result in good performance on high-resource languages, but poor performance on the low-resource languages. Sampling equally from all language pairs would result in huge improvement in low-resource translation performance, but high-resource languages perform significantly worse than their bilingual baselines.

To balance between high and low-resource language pairs, we used the temperature-based sampling strategy used for training multilingual BERT [4] [7]. For a given language pair, $l$, let $D_l$ be the size of the available parallel corpus. If we sample from the union of the datasets, the probability of the sample being from language $l$ is $p_l = \frac{D_l}{\Sigma_l D_l}$. We set the probability of our sampled distribution to be proportional to $p_l^{\frac{1}{T}}$, where $T$ is the sampling temperature. Now, $T = 1$ corresponds to true data distribution and $T = 100$ corresponds to an (almost) equal number of samples for each language. We used $T = 5$ for our multilingual model.

For all our multilingual experiments, we trained a single Transformer model simultaneously on all languages, with the same hyperparameters as the bilingual model. We used a shared SPM vocabulary with 64K tokens, generated using the same sampling distribution ($T = 5$) used during training. We additionally used character coverage of $0.999995$ to ensure our vocabulary contained most of the alphabets for all 103 languages.

## 3.3 Effects of large batch size

Due to its simplicity, data parallelism is the dominant approach to scale neural network training[8, 9]. We test the limits of large-batch training by significantly increasing the batch size used for standard Transformer Big training. Starting from 260K tokens per batch, we increase the effective batch size to 4M and observe the validation loss and BLEU scores on the high-resource language pair, German-English (similar trend can also be observed for other language pairs). Optimization parameters used here are identical to those for previous experiments. To our knowledge, 4M tokens per batch is the largest batch size that has ever been used in literature to date for training NMT models [10]. Table 2 shows that both metrics improve significantly as we increase the batch size. We believe further increasing batch size can potentially yield more improvement.

Table 2: The Effect of Batch Size

| Batch Size | 260K | 1M | 4M |
|---|---|---|---|
| BLEU | 30.92 | 31.86 | 32.71 |
| Loss (NLL) | 2.58 | 2.51 | 2.46 |

## 4 Discussion

Having a flexible framework for large-scale deep learning experiments opens up exciting opportunities to understand the underlying machinery and mechanics of large-scale models. In this section we relate our experimental results with some recent studies in deep learning. We also share some additional empirical findings of interest with deep learning practitioners.

**Expressivity and Generalization:** Recent findings in deep learning theory [11, 12] postulate increased generalization performance with growth in the expressive power of the network.[5] Here we present our experiments toward an empirical validation. We increase the depth as a means to increasing the network's expressive power [13] while controlling the batch size. We are able to observe the generalization behavior of the network at a scale that has never been experimented with before. Starting from a 6-layer Transformer Big (12 layers in total with encoder + decoder), we gradually increase the depth up to 64 layers (128 layers in total). We observe that the 64-layer model exhibits almost an up-shifted trend of the 6-layer model and all the intermediate depths lie in between. Although our results support the theory, we also observe diminishing returns, which

raises trainability concerns.We may not yet have the tools/techniques to reduce the training error even further, highlighting the necessity of understanding trainability challenges for further progress.

**Depth-Width Trade-off:** Another area that has attracted the attention of deep learning theoreticians is the effect of model width and depth on generalization [14, 15]. We next study the trade-off between depth and width in our multilingual setup and compare the performance of 1.3B wide, $T(12, 16384, 32)$, and 1.3B deep models, $T(24, 8192, 16)$. While the performance of these two models on high-resource languages is very similar, the deeper model outperforms low-resource languages by huge margins, suggesting that increasing model depth might be better for generalization.

Figure 3: The Effect of Depth

Further, the performance improvement for low-resource languages (right side of Figure 3), when comparing the 1.3B deep model against the 400M model, is almost as large as the improvement for high-resource languages, indicating that increasing depth might also increase the extent of transfer to low-resource tasks.

**Faster Convergence with Depth:** We report one intriguing observation correlated with making our models deeper. Keeping the effective batch size, optimizer hyperparameters and model width fixed, increasing model depth induces speedup in optimization, as illustrated in Figure 3. Previous work reporting similar phenomena conjectured that depth induces pre-conditioning by over-parametrization (Figure 4 in [16]).

While this paper focuses primarily on systems challenges associated with scaling neural networks, there is a growing need for theoretical understanding of deep neural networks in order to better address generalization and trainability concerns. We believe flexible scaling tools such as GPipe are essential in bridging the gap between deep learning theory and practice. We hope that our empirical findings and discussions will motivate more research efforts along this path.

## Footnotes

[1] https://github.com/stanford-futuredata/dawn-bench-entries/blob/master/ImageNet/train/google_amoeba_net_d_tpu_tensorflow18.json

[2] (3.0, 40K) indicates a schedule with a learning rate of 3.0 and 40K warm-up steps, which is decayed with the inverse square root of the number of training steps after warm-up.

[3] We used an in-house implementation mteval-v13a.pl from Moses to evaluate BLEU scores for our multilingual experiments.

[4]https://github.com/google-research/bert/

[5]In other words, both estimation error and approximation error decrease when we expand the hypothesis space - enlarging the models in practice, indicating that there is no bias variance trade-off in the classical sense.