[Reviews · NeurIPS 2019]

Reviewer 1



Originality * Their proposed algorithm has little in the way of surprising conceptual insights, but in this case that is a good thing - the parallelism algorithm is simple, intuitive, and achieves nearly linear throughput increase in the number of accelerators used (hard to expect much more). It's surprising that this hasn't been done before, given that it's such a simple trick that gives such a large speed up for training distributed models. * The authors propose a couple smaller tricks to make the parallelism algorithm work or work better (re-computing forward pass activations to reduce memory requirements, doing so while waiting for backward pass gradients). To train their very deep models, the authors also use another few smaller tricks (e.g., clipping logits to mitigate bad gradients) Significance & Quality * General-purpose model parallelism algorithm: The proposed algorithm is applicable to almost any neural network architecture without modification; the authors demonstrate this feature by scaling up state-of-the-art architectures in both computer vision and NLP settings. * Empirical results with scaled up models: The large-scale models enabled by TensorPipe show empirical gains on highly competitive benchmarks in computer vision (ImageNet) and NLP (machine translation). In machine translation for low-resource languages, these gains seem quite substantial. Clarity: Clearly written, aided by the simplicity of the algorithm. Figure 2 is a clear, comprehensive overview of the approach. The writing could be made more concise/dense in some places, especially to make more space for the helpful analysis of wall-clock time breakdown referenced in the Appendix (2.2). The authors clearly describe the relation of their work to other parallelism algorithms, plus the broader literature regarding deeper/wider models and generalization. The authors may also wish to relate to the literature in developing model architectures that are explicitly easy to distribute (e.g. "Outrageously Large Neural Networks: The Sparsely-Gated Mixture-of-Experts Layer"). It also may be helpful to briefly overview (or show in a figure) the architectures for AmoebaNet and Transformer.

Reviewer 2



There exists a strong correlation between model size an accuracy on large benchmark datasets. Accordingly, it is valuable to push the limits of scale and see if/when the gains from increasing model size saturates. One of the leading approaches for scaling model sizes is model parallelism, but model parallel training is complex and often much slower than standard training. The authors propose a framework for combining *pipeline* model parallelism with gradient checkpointing ("re-materialization"). Compared to other model parallel approaches that split individual layers across accelerators, the proposed TensorPipe approach partitions whole layers across devices and uses pipelining to improve overall efficiency and gradient checkpointing to reduce inter-device communication. The proposed approach works, and the authors demonstrate strong results scaling model sizes by an order of magnitude for image classification and machine translation models. The paper is well written, the code is open source, and the results are compelling. This work is likely to have a large influence on future model scaling efforts and serves as a good reference for future adoption/implementation in other frameworks.

Reviewer 3



The article is well-written and easy to read. The article addresses an important challenge: achieving model parallelism without being architecture and task specific. The solution is simple and straightforward. The key contribution pertains to the implementation and empirical evaluation of the system. The scheduling algorithm should be elaborated. It is not clear how the load imbalance is addressed and how possible node failures are taken into account. The minibatching algorithm details are not presented in the paper. The evaluation does not include a benchmark system. The major limitations of the article pertain to the missing algorithm detail pertaining to the scheduling/microbatching and the lack of empirical benchmark system. The system configuration parameters are explored with the empirical setup. The system has been contributed to Open Source. Author response: the authors have addressed the above points by providing more explanation of the scheduler that is also open sourced and more details on the micro batching algorithm.

[Author Response · NeurIPS 2019]

¹ We thank the reviewers for the generous comments and invaluable feedback on the manuscript. Please find below our
² point-by-point response to reviewers' concerns.

³ **Reviewer #1**

⁴ • *"Analysis of wall-clock time breakdown"*:
⁵ We will move the wall-clock time breakdown analysis into the main text of the paper.

⁶ • *"Literature of architectures that are explicitly easy to distribute"*:
⁷ Thank you for pointing out the related work, which we will cite in the camera-ready version. We would also
⁸ like to emphasize that although mixture-of-experts-type architectures are relatively easy for device placement,
⁹ the dispatching of examples and gathering of experts outputs is often non-trivial, and comes at the cost of
¹⁰ expert load balancing, expert utilization, batch-size tuning etc. We will add a detailed discussion regarding this
¹¹ in the camera-ready version.

¹² • *"Overview the architectures for AmoebaNet and Transformer"*:
¹³ We will include a brief overview and figures for the architectures considered here in the appendix.

¹⁴ • *"Suggestions on improvements"*:
¹⁵ Thank you very much for the suggestions. We have been running experiments with very deep transformer
¹⁶ language models and are currently exploring approaches to stabilize training for these models, beyond those
¹⁷ discussed in the paper. We will try to share our findings in the camera-ready version. At the same time, we
¹⁸ will soon open source the code to train 128+ block transformers on Github to allow researchers to train these
¹⁹ models with their own data.

²⁰ **Reviewer #2**

²¹ • *"Batch size experiments feel a bit out of place"*:
²² Thank you for the suggestion. We added this discussion in order to provide more details for our machine
²³ translation experiments. We will move this set of results into the supplementary material and move the
²⁴ wall-clock time breakdown analysis into the main paper as suggested by reviewer #1.

²⁵ **Reviewer #3**

²⁶ • *"Elaborate the scheduling algorithm"*:
²⁷ We agree with the reviewer that we should have done a better job at explaining our algorithms in greater
²⁸ details. Currently, we allow the user to specify the per-layer computation cost for each layer ($c_i$) as measured
²⁹ in flops, as briefly mentioned in Section 2.1. For distributing layers across accelerators we try to balance the
³⁰ per-accelerator computation cost by minimizing the variance in the estimated cost of all accelerators. We have
³¹ open sourced our heuristic algorithm in our open-source repository. Since actual computational time might
³² differ from the flops estimation $c_i$, depending on different accelerators and architectures, we also allow the
³³ user to manually specify layer placement. We are also planning to further improve our scheduling algorithm
³⁴ to balance both memory utilization and computational time in order to maximize batch sizes and training
³⁵ efficiency. We will describe the scheduling algorithm more clearly in Section 2 in the camera-ready version.

³⁶ • *"How possible node failures are taken into account"*:
³⁷ Our system uses synchronous training so any one of the machine failures would force the whole system to
³⁸ restart from the previous checkpoint, as with standard non-model parallelism training.

³⁹ • *"The micro-batching algorithm details are not presented in the paper."*:
⁴⁰ The overview of the micro-batch algorithm is depicted in Figure 2(c). We first split the tensor of mini-batch
⁴¹ input along the batch dimension into micro-batches and apply control-flow ops to loop over each micro-batch
⁴² to compute the gradients. Accumulated gradients over all micro-batches are applied to variables at the end of
⁴³ each global step. We will describe the micro-batching algorithm more clearly in Section 2 of camera-ready.

⁴⁴ • *"The lack of empirical benchmark system."*:
⁴⁵ We agree that providing a unified benchmark system for scaling giant neural network would be beneficial to
⁴⁶ the research community. However, we are afraid it is a bit beyond the scope of our paper to define such a
⁴⁷ benchmark system to evaluate all model parallelism libraries. For example, Mesh TensorFlow does not support
⁴⁸ convolution operations, making it infeasible for us to compare our image model performance with theirs.

⁴⁹ • *"Missing reference in the supplementary material"*:
⁵⁰ Thank you for pointing this out. We will fix this in the camera-ready version.

[Meta-Review · NeurIPS 2019]

While the proposed pipelining framework is simple, the reviewers unanimously assert that from an empirical point of view it is an important advance, especially in language modeling tasks.